REGISTERED REPORT PROTOCOL

# Protocol for a longitudinal study investigating the role of anxiety on academic outcomes in children on the autism spectrum

**Dawn Adams**[1]*, **Stephanie Malone**[1], **Kate Simpson**[1], **Madonna Tucker**[2], **Ron M. Rapee**[3], **Jacqui Rodgers**[4], **Deb Keen**[1]

**1** Autism Centre of Excellence, Griffith University, Mount Gravatt, Queensland, Australia, **2** AEIOU Foundation, Brisbane, Queensland, Australia, **3** Department of Psychology, Centre for Emotional Health, Macquarie University, Sydney, Australia, **4** Population Health Sciences Institute, Newcastle University, Newcastle Upon Tyne, United Kingdom

* dawn.adams@griffith.edu.au

## Abstract

### Background

Children on the autism spectrum are consistently reported to underachieve compared to ability. In typically developing children, anxiety is a strong predictor of poor school performance. Despite the high prevalence of anxiety disorders among children on the spectrum, the impact of their anxiety on academic achievement is under-researched. The main aim of this project is to determine the moderating role anxiety may have in the development of academic learning behaviours (academic enablers) in children on the spectrum. This project addresses a gap in knowledge about the possible associations between anxiety and academic achievement in children on the spectrum. Understanding these associations opens up the possibility of new intervention pathways to enhance academic outcomes through anxiety reduction/prevention.

### Methods

This longitudinal study will aim to recruit 64 children on the spectrum aged 4–5 years and their parents. Information will be gathered from children, parents and teachers. Children will be randomly assigned to one of two conditions in order to experimentally manipulate anxiety levels in the sample: experimental (to receive an anxiety reduction/prevention program, $N = 32$) or control (no intervention/treatment as usual, $N = 32$). The primary outcome measures are child academic skills and enabling behaviours assessed using the Academic Competence Evaluation Scales and the WIAT-II. Anxiety will be assessed through parent and teacher report. Assessments will be conducted at baseline, post-experimental manipulation of anxiety, and within the first year of formal schooling. It is hypothesised that anxiety will moderate the relationship between autism characteristics and academic enablers.

### Dissemination

Results will be disseminated through peer-reviewed manuscripts and conference presentations. Lay summaries will be provided to all participants and available on the research centre website.

This is a Registered Report and may have an associated publication; please check the article page on the journal site for any related articles.

**Data Availability Statement:** The anonymised dataset for the findings of the study outlined in this Registered Report Protocol will be stored in Griffith's Research Data Repository, which is accessible and searchable through a web interface (https://www.griffith.edu.au/library/research-publishing/repository).

**Funding:** This study is funded by a Linkage Project grant, awarded to Griffith University and AEIOU Foundation by the Australian Research Council (Grant No.: LP180100318). The Linkage Project scheme recognises collaborative partnerships between research and industry, fostering the transfer of knowledge, skills and ideas between sectors (see https://www.arc.gov.au/grants/linkage-program/linkage-projects). Neither the funder, nor any of the employing Universities of the investigators, have influenced the study design, data collection, or will influence the decision to submit the final results for publication.

**Competing interests:** Professor Ron Rapee developed the intervention (Cool Little Kids) that is being used in the study, but other than that, there are no known conflicts of interest associated with this publication. This does not alter our adherence to PLOS ONE policies on sharing data and materials.

# Introduction

Children on the autism spectrum are consistently reported to underachieve academically compared to their ability [1–3]. These levels of underachievement remain present into adulthood [4], and are associated with reduced social and vocational outcomes [5]. Academic underachievement is understudied in autism, despite its high prevalence and potentially long-lasting impact. Consequently, little is known about associated factors [6]. In a review of 19 studies investigating academic achievement in autism, Keen *et al.* identified no consistent predictors [7]. Since this was published, there have been some studies linking previously unexplored factors, such as some elements of executive functioning, to academic attainment [8,9]. Notable from the work to date is the lack of research on academic enablers: attitudes and behaviours that facilitate students' participation in and benefit from academic instruction in the classroom' [10].

Academic success relies on a combination of academic skills and enablers that include engagement, motivation, study skills, and social skills [11]. Our pilot research has shown that these enablers are poorer in students on the spectrum compared to children without autism. Critically, in children on the spectrum, these academic enablers were stronger predictors of academic achievement than language skills, even in children as young as 5 years old [12]. It is not clear, however, why students on the spectrum have poorer scores on academic enablers, or whether it is possible to enhance these to improve academic outcomes. One plausible hypothesis is that anxiety (a common co-occurring experience in autism) is impacting on or possibly interacting with the child's academic enablers and their ability to engage and participate in education, making it more difficult for students to benefit from academic instruction in the classroom and contributing to poorer academic achievement.

Anxiety disorders are the most common form of mental disorder in young people, affecting around 7% of Australian youth [13]. Anxiety disorders are even more common among children on the spectrum, affecting up to 40% [14] with an even larger number showing sub-clinical levels of elevated anxiety [15]. When researching anxiety in autism, it is important to consider both those who meet criteria for an anxiety disorder and those at the sub-clinical level where anxiety can still impact day-to-day functioning. Heightened levels of anxiety have been reported among children on the spectrum as young as 5–6 years of age [16]. These heightened anxiety levels have been identified as a significant predictor of educational quality of life using both parent and child self-report [17,18], and also rated by teachers and parents as one of the factors having the most impact on the educational support needs of students on the spectrum [19].

Among typically developing children, anxiety is a strong predictor of poor school engagement and performance [20]. Anxiety has been linked to below-grade-level academic achievement, school failure, and academic skill impairment [21]. Subsequently, reducing anxiety has led to improved school engagement and performance [20]. Based on these relations, the dramatic impact of autism on academic achievement may be at least partly a consequence of the high levels of associated anxiety.

Given that both academic outcomes and academic enablers are poorer in children on the spectrum, it is reasonable to hypothesise that the poor academic success noted within this population is likely a consequence of the impact of autism on the development and/or implementation of academic enablers. It is also highly likely that anxiety is impacting on these same enablers. For example, extensive evidence with older, typically developing children has demonstrated the impact of anxiety on social performance and interactions [22]. Similarly, perfectionism and worry, which are key components of anxiety, can have a negative impact on study skills and motivation [23]. At a broader level, enablers of good academic functioning often

reflect the ability to focus attention and think flexibly. A core mechanism of anxiety, threat expectancy, interferes with a range of cognitive processes, including attentional focus [24]. In addition to heightened threat expectancy, an additional key mechanism in the maintenance of anxiety is a reduced ability to tolerate uncertainty (intolerance of uncertainty [IU]) [25]. High levels of IU are likely to be especially pertinent to children on the spectrum. Many of the core symptoms of autism reflect a need for consistency and predictability, suggesting that threats to consistency and predictability are likely to be highly distressing to these children. Several studies have shown high levels of IU among children, adolescents, and adults on the spectrum [26,27]. Additionally, IU has been found to mediate the relation between autism and anxiety [28], where higher levels of IU are associated with higher levels of anxious symptomatology in these individuals. IU has also been shown to predict parent and family responses to anxiety management [29,30] as well as levels of school functioning using parent and child reports, including those specific to quality of life in children on the spectrum [17,18,31,32].

In summary, it is likely that heightened anxiety, at least partially, interacts with academic enablers to contribute to the poor academic performance consistently reported for children on the spectrum. However, to date, this hypothesis has not been tested. The primary aim of this study is therefore to evaluate the role of anxiety on academic outcomes (academic skills, academic enablers) in children on the autism spectrum. The most valid way to determine causal status (i.e. that anxiety impacts upon academic outcomes) is through experimental manipulation. Therefore, this study will experimentally manipulate anxiety (through an intervention) to determine whether this manipulation impacts academic enablers and outcomes. The case for needing to experimentally manipulate anxiety is strengthend by the findings that that 70–80% of children on the autism spectrum experience elevated anxiety levels and40% of children on the spectrum have a clinical diagnosis of an anxiety disorder [14]. Therefore, experimental manipulation which aims to prevent or reduce anxiety increases the opportunity for a broader range of anxiety scores within the sample. An extensive body of literature has demonstrated the efficacy of skills-based intervention programs to reduce heightened anxiety among typically developing children as young as 3 years of age [33]. One such program is Cool Little Kids (CLK), a six-session, parent-led program that has demonstrated excellent outcomes in the reduction of anxiety among preschool-aged children. The core treatment strategy in this program is in vivo exposure, which primarily works via extinction to shift expectations of threat [34]. In a recent analysis, Bischof *et al* showed that CLK also led to reductions in anxiety among a group of preschool-aged children with co-occurring autism, with feedback from parents suggesting that future offerings of the intervention could use examples and information that was more autism focused [35].

Although CLK has shown promise in reducing anxiety among young children on the spectrum, its processes focus primarily on one mechanism of anxiety, threat expectancy. Given the elevated levels of both IU and anxiety noted in children, adolescents and adults on the spectrum [36,37], more complete reduction in anxiety might be achieved by also reducing IU. Rodgers *et al* [38,39] recently developed a parent-mediated intervention especially for children (aged 8–15 years) on the spectrum (Coping with Uncertainty in Everyday Situations; CUES) that has shown promising efficacy [38]. Therefore, combining the processes of CLK and CUES (hereafter termed CLK-CUES) should allow greater reduction in anxiety among children on the spectrum, leading to a stronger experimental manipulation of the critical variable.

## Aim and hypotheses

The aim of this project is to determine the role that anxiety may have in the development of academic learning behaviours (academic enablers) in children on the spectrum. Experimental

manipulations are the only way to determine causal status. Therefore, we will experimentally manipulate anxiety (through the CLK-CUES intervention) to determine if this manipulation affects academic outcomes at follow-up. This methodology will allow for the following research questions to be posed:

1. Is there evidence that anxiety is associated with academic enablers in children on the autism spectrum?

2. Does anxiety moderate the relationship between autism characteristics and academic enablers in these children?

The answers to these questions will inform a model for intervention aimed at children on the spectrum entering their first year of formal schooling. It is hypothesised that anxiety will moderate the relationship between autism characteristics and academic enablers and skills. Additional measures of factors associated with anxiety and/or academic achievement, such as sensory profiles and repetitive behaviours, are included so they can be accounted for in secondary analyses if required.

## Methods and analyses

### Overview of method

This project uses a longitudinal design with participants participating in three data collection points over 2 years: baseline (Time 1; T1); post-completion of the CLK-CUES program or the equivalent number of weeks after baseline for the control group (Time 2; T2, to check the efficacy of the experimental manipulation of anxiety); and during the first year of formal schooling (Time 3; T3).

Parents, and their children on the spectrum (4- to 5-year-olds), will be recruited the year prior to their child entering formal schooling. As shown in Fig 1, half of the parents will be randomised to receive the CLK-CUES program while the remaining half will form the control group. Randomisation of participants into these two groups ensures the randomisation of potentially confounding factors across the levels of anxiety. Groups will be run during school-terms to reduce any barriers relating to childcare. In order to minimise the risk of participant drop out between randomisation and commencement of the intervention, recruitment advertising will be timed to enable recruitment, initial assessment and randomisation occurs as close to the onset of the groups as possible whilst also allowing parents sufficient notice of group onset. Final assessments will be conducted within 6 months of children transitioning to formal schooling. Parents and children will complete measures at all time points (T1-T3), with additional data collected from the child's schoolteacher at T3.

### Trial registration

The primary aim of the study is to explore the role of anxiety on academic outcomes for young children on the autism spectrum. To determine causal status, an experimental manipulation of anxiety is requiresd. Therefore, some parents will receive a modified version of an intervention shown to have efficacy in reducing anxiety in neurotypical children and those on the autism spectrum [33] and others will not. Allocation into the intervention or control group will be randomised. The primary aim of the study is to explore the impact of anxiety on academic outcomes, but as assumption behind the experimental manipulation (that the intervention reduces anxiety) will require a comparison of the two groups on anxiety levels at Time 2 and 3, the study will be able to test the efficacy of this intervention to prevent or reduce anxiety. Therefore the study is registered with Australian New Zealand Clinical Trials Registry

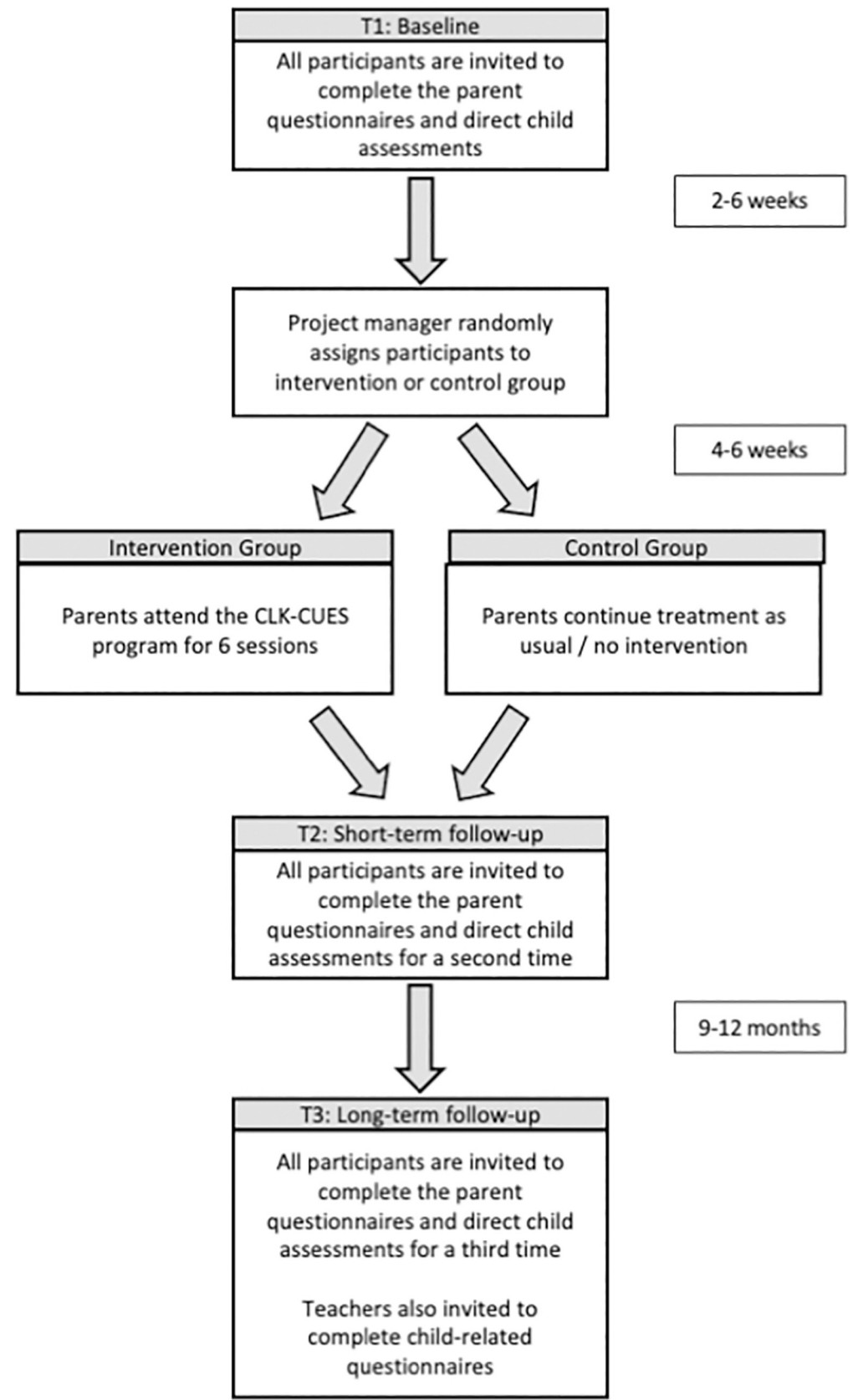

**Fig 1. Flow diagram for study.**

(ANZCTR) number ACTRN12620001322921. To ensure clear reporting of this protocol, the SPIRIT checklist has been used.

## Recruitment and participants

As there has been no previous research exploring the effect of anxiety on academic outcomes for children on the autism spectrum, the target sample size was informed by a power analysis (using Gpower [40]) and determined through analysis of previous research on anxiety and academic outcomes in neurotypical students. In a large study of 715 elementary school students, Xu, Huebner and Tian (2021) report an effect size of anxiety on academic outcomes (when comparing high and low anxiety groups) of $d = 0.68$. Using this effect size in Gpower, the sample size required to conduct the linear regression with four predictor variables (autism severity, ability, anxiety and the interaction of autism and anxiety) on academic outcomes with a significant level of 5% and 80% power to reject the null hypothesis of no difference is 42.

A second power analysis was undertaken to ensure that there is a significant power to test for the effectiveness of the experimental manipulation. The effect size from Bischof *et al* [35] who piloted the original CLK program with children on the spectrum by evaluating outcomes within a larger population trial of 13 parents of children on the spectrum in the intervention group and 13 in the control group, was used. Bischof *et al*. reported an effect size of Cohen's $d = .91$ on the anxiety measure (Preschool Anxiety Scale-Revised; PAS-R) [41] at 2-year follow up. Conservatively assuming a more modest effect (Cohen's $d = .75$) and applying a retention rate of 90% [35] and a significance level of 5%, it is estimated that 64 participants (32 in each group) will be needed to obtain a statistical power of 80% to reject the null hypothesis of no difference. This calculation is based on the $t$ statistic.

Based on these two power analyses and their assumptions, in order to ensure an accurate test of the experimental manipulation (i.e. a sufficient sample size to evaluate if the intervention resulted in a significant difference between the two groups on level of anxiety) and ensure a sufficient power in the linear regression to answer the reseach questions, 64 participants will be enrolled. The 64 parents will be recruited through social media and AEIOU autism early learning centres. If there is a shortfall of participants, other early intervention service providers in Queensland will be approached. Inclusion criteria state that parents will be eligible to participate if their child is aged 4–5 years, has a score indicating autism spectrum on the Autism Diagnostic Observation Schedule-2 (ADOS-2) [42], and has not yet commenced their first year of formal schooling and are able to attend the parent group sessions. There are no set exclusion criteria.

Recruitment will occur in waves to assist in meeting recruitment targets and to ensure sufficient available resources to implement the intervention with 32 parents. Each recruitment wave will comprise of 12–20 families. For each wave, the project manager will use a random number generator, to randomly allocate half of the families to the experimental condition (CLK-CUES anxiety prevention program) and half to the control condition (no intervention/ treatment as usual). Treatment as usual was selected as the comparator so as to allow for a broad range of anxiety levels across the two groups, and to indicate what would have been the common outcome had the intervention not been implemented. The project manager will notify parents of their group status via letter/email and will be asked to select the workshop location and time most appropriate for them. This project manager will also maintain a record of participants allocated to the conditions. The research team members responsible for collecting the direct pre- and post-intervention measures with the children will be blind to group allocation of participants and teachers will not be informed of which condition the child was in. It is not possible for parents or clinicians implementing the intervention to be blinded to group status, however, the clinicians are not involved in the data collection.

At T3, teachers of the participating children will be recruited to obtain their perspectives on the child's school-related anxiety and academic competence. These teachers will be blind to the child's group allocation. Given the longitudinal nature of this research, incentives (gift cards) will be offered to parents following data collection and free day of professional development on autism in schools offered to participating teachers following the end of data collection.

## Intervention used to experimentally manipulate anxiety levels

The CLK-CUES intervention contains the standard information from the manualised CLK parenting program [43] with the addition of specific information from the CUES intervention as noted below. CLK is a parent-mediated, group-based intervention in which parents of children aged 3–6 attend six sessions (one per week) of 2 hours duration. In the current study, each group will consist of four to eight parents. If parents wish to discontinue the intervention, they are able to stop attending at any point.

As mentioned previously, CLK focuses on threat extinction and has been empirically validated for 3- to 6-year-olds at risk of developing anxiety [44]. To adapt this intervention for children on the spectrum, the research team have modified the clinician manual and parent workbook to include autism-specific content. This modified version contains all of the elements of the original CLK program but also draws upon the CUES intervention as well as broader knowledge related to anxiety in autism to incorporate autism relevant topics and strategies which address IU [38]. A pilot study (*n* = 3 parents) has provided positive feedback on the use of this modified program for young children on the spectrum. For more details on the intervention content, please contact the lead author.

The CLK-CUES program will be administered by a trained member of the research team and delivered in community settings across Queensland, Australia. Intervention fidelity will be assessed by the facilitator during session delivery using a purpose-designed content checklist. This same checklist will also be used by a second researcher to assess the fidelity of 20% of the sessions (using audio recordings of the sessions), this researcher will not know the identies of the group members, nor will they be involved in any of the direct assessments of the children or parents. Taken together, this will allow for inter-rater reliability to be established. Parent attendance and homework implementation will be recorded as measures of adherence.

## Measures

In addition to collecting general demographic information and, at Time 2 and 3, information on any supports and services for anxiety that have been received since the last assessment, a range of child-directed, parent and teacher measures will be collected, these are presented in Tables 1–3, respectively. All parent/teacher questionnaires are administered via an online survey provider (REDCap). The order of the questionnaires will be set so that parents will be first asked to complete variables in the primary analyses before those required for secondary or exploratory analyses, to minimise the risk of any missing data on the trial outcomes. Child-directed assessments are administered in a 1:1 setting by a member of the research team. The two primary outcomes will related to academics; (1) academic enablers measured by the Academic Competence Evaluation Scales—Teacher Form (ACES-TF) [11] and, to minimise the impact of missing teacher data, (2) the Wechsler Individual Attainment Test (WIAT-III).

Age-appropriate subscales from a teacher-completed anxiety measure, the Preschool Anxiety Scale Revised (PAS-R) [39], and the parent-completed Anxiety Scale for Children—Autism Spectrum Disorder (ASC-ASD) will be collected so as to gain a quantifiable measure of anxiety [50]. As part of the intake process at AEIOU centres, the majority of children complete the

**Table 1. Child assessments used in study.**

| Construct | Measure | Administration point | Reliability | Additional information |
|---|---|---|---|---|
| Autism characteristics | Autism Diagnostic Observation Schedule 2nd Edition (ADOS-2) [42] | T1, T2, T3 | Test-retest reliability: $r$s = .83 - .87 Inter-rater reliability: $r$s = .94 - .97 [43] | Although originally designed as a diagnostic assessment, the ADOS-2 is commonly used in research as a quantitative measurement of autism |
| Cognitive ability | Mullens Scale of Early Learning (MSEL) [45] | T1, T2, T3 | Internal consistency: $\alpha$s = .75 - .83 [45] | Well-standardised measure of ability for children aged 0–68 months. Provides a measure of child ability (visual reception, receptive language, expressive language, gross motor and fine motor) and a Developmental Quotient (DQ). Commonly used as a descriptive and outcome measure for children on the spectrum [46]. |
| Academic performance | Wechsler Individual Achievement Test—Australian and New Zealand Standardised, 3rd Edition (WIAT-III) [47] | T3 | Split-half reliability: $r$s = .69 - .98 [47] | **Primary outcome measure 1** Standardised measure of academic performance. For children aged 5 years, it provides assessments of alphabet writing, spelling, oral expression, listening comprehension, early reading skills, mathematics and numeracy |
| School connectedness | To be developed across duration of study | T3 | n/a | This will need to be specifically developed as there are no measures of school connectedness for this age (see [48]) |

Note. T1 = Baseline; T2 = Post-intervention; T3 = 1-year follow-up.

ADOS-2 and Mullen Scales of Early Learning (MSEL) [45] assessments, and parents provide information on their child's daily functioning (Vineland Adaptive Behavior Scales-3) [54]. To save repeating these assessments at T1, parents recruited via AEIOU centres will be asked to consent for these data to be shared with the research team. No restrictions are placed upon parents accessing any other interventions or services during their participation in the study, but they will be asked to note any services accessed since the previous assessment during their T2 and T3 questionnaire completion. Additional measures of variables which may impact outcomes (sensory profiles, impact of anxiety, restricted and repetitive behaviour) are included for secondary exploratory analyses, informed by the outcomes of the primary analysis.

## Data analysis

Missing data that are deemed "missing at random" within a timepoint will be imputed as determined by the manual for each measure. Missing timepoints will be approached using intention to treat analysis. Scoring of all questionnaires completed on the online software is through syntax, minimising the risk of data entry errors, However, prior to any analysis, range checks and descriptive statistics will be undertaken to identify any data entry errors. Then, a manipulation check of anxiety levels at T2 and T3 will be undertaken to ensure the short and longer-term experimental manipulation of anxiety (through the parent intervention) was successful. Without this, causal status between anxiety and academic outcomes cannot be determined. This will be completed through a t-test with the independent variable of condition (received CLK-CUES intervention vs. control) and the dependent variable child anxiety levels.

To answer the first research question, and to determine whether anxiety is associated with academic outcomes in children on the spectrum, linear regression analyses will be conducted for each academic outcome. The primary variable of interest is the ACES academic enablers total score and subscales (Interpersonal skills, engagement and motivation), a teacher-completed questionnaire collected at T3. To reduce the risk of relying solely on teacher data, child assessments of academic competence will also be undertaken at T3. To investigate the variance

**Table 2. Parent measures used in study.**

| Construct | Measure | Administration point | Reliability | Items (number and response procedure) | Additional information |
|---|---|---|---|---|---|
| Autism characteristics | Social Communication Questionnaire-Lifetime (SCQ) [49] | T1, T2, T3 | Int consist: α = .84-.93 | 40 items (yes/no) | Assesses parent(s) perspective on their child's autism characteristics Validated in children aged ≥4y [49] |
| Child anxiety | Preschool Anxiety Scale Revised (PAS-R) [41] | T1, T2, T3 | Int consist: α = .72-.92 12-month stability: rs = .60 - .75 | 34 items (0 = not true at all; 4 = very often true) | **Secondary outcome measure** Measures generalised, social and separation anxiety, and specific fears in children < 6 years [41] |
|  | Anxiety Scale for Children —Autism Spectrum Disorder (ASC-ASD) [50] | T1, T2, T3 | Int consist: α = .94 test-retest reliability: r = .84 [50] | 24 items (1 = never; 4 = often) | **Secondary outcome measure** Autism-specific of anxiety in children ≥ 5y in descriptive and intervention studies [38] |
| Child intolerance of uncertainty | Responses to Uncertainty and Low Environmental Structure (RULES) [51] | T1, T2, T3 | Int consist: α = .93 | 17 items (1 = not at all, 5 = very much) | Developed to measure need for rules and certainty for children aged 3–10 years [51] |
|  | Intolerance of Uncertainty Scale–Parent (IUS-P) [28] | T1, T2, T3 | Int consist: α = .90 (ASD) and .91 (TD) | 12-items (1 = not at all; 5 = very much) | Assesses children's emotional, cognitive and behavioural response to IU [28] |
| Impact of child anxiety | Child Anxiety Life Interference Scale Preschool (CALIS-P) [52] | T1, T2, T3 | Int consist: ω = .88 [53] | 18 items 5-point scale ("not at all" to "a great deal") | Designed for children aged 3 to 5years to assess life interference attributed to fears and worries |
| Child adaptive behaviour | Vineland Adaptive Behavior Scales 3rd Edition (VABS-3) [54] | T1, T2, T3 | Adaptive behavior composite Int consist: α = .97 | 99 items (communication) 109 items daily living 99 items socialization 76 items motor skills (0 = never; 2 = usually/) | Used extensively to measure adaptive behaviour of children on the spectrum. |
| Child repetitive behaviour | Repetitive Behaviour Scale —Early Childhood (RBS-EC) [55] | T1, T2, T3 | Int consist: α = .70-.3 Test-retest reliability: ICC = .87 | 34 items (0 = does not occur to 4 = occurs many times a day) | Developed for children from infancy through early school age [55] |
| Child sensory profile | Short Sensory Profile 2 (SSP-2) [56] | T1, T2, T3 | Int consist: α = .57-.92 Test-retest reliability: rs = .83 - .92 [56] | 34 items 6-point scale (5 = almost always [>90% of time]; 0 = does not apply [have not observed the behaviour]) | Measures behaviours associated with abnormal responses to sensory stimuli, and has been used successfully with children on the spectrum |
| Parental mental health | Depression Anxiety Stress Scales (DASS-21) [57] | T1, T2, T3 | Int consist: α = .88 [58] | 21-items (0 = did not apply to me at all; 3 = applied to me very much or most of the time) | Widely used in clinical and non-clinical samples, including parents of children on the spectrum [59] |
| School attendance | School Non-Attendance ChecKlist (SNACK) [60] | T3 | n/a | Parents indicate which of 14 reasons best accounts for each school absence. If none is appropriate, they are asked to briefly describe the reason | Combines a vast literature on school attendance problems; screens for the four main categories of absenteeism (school withdrawal, school exclusion, truancy and school refusal) and non-problematic absenteeism [60] |

*Note.* T1 = Baseline; T2 = Post-intervention; T3 = 1-year follow-up. Int. consist = Internal consistency.

in academic outcomes explained by anxiety when controlling for autism severity (ADOS score) and cognitive ability (DQ or adaptive behaviour), two linear regressions will be undertaken, one focussing upon teacher report and the other on parent report of anxiety. There will be four predictor variables in each linear regression; ADOS score, measures of ability, a measure of anxiety (teacher or parent) and the interaction of anxiety and ADOS score. One linear regression will have the outcome variable of the ACES score and one the WIAT score. The

**Table 3. Teacher measures used in study.**

| Construct | Measure | Administration point | Reliability | Items (number and response procedure) |
|---|---|---|---|---|
| Child academic enablers | Academic Competence Evaluation Scales—Teacher Form (ACES-TF) [11] | T3 | Internal consistency: $\alpha$ = .94 - .99 Test-retest reliability: $r$s = .88 - .97 [11] | **Primary outcome measure 2** 73 items: 33 on ability and importance of academic skills (1 = far below expectations; 5 = far above) (1 = not important; 3 = critical) 40 on frequency and importance of academic enablers (1 = never; 5 = almost always) (1 = not important; 3 = critical) |
| Child school anxiety | School Anxiety Scale—Teacher Rating Scale (SAS-TR) [61] | T3 | Internal consistency: $\alpha$ = .93 Test-retest reliability: ICC = .70–92 [61] | Teachers will provide an indication of how often the child has demonstrated 16 school-related anxiety behaviours in the past 3 months (0 = never; 3 = always) |

inclusion of the interaction between the anxiety and ADOS score will be used to answer the second research question and to determine whether anxiety moderates the relationship between autism and academic enablers or outcomes.

## Methodological considerations

**Bias in recruitment.** This research project requires a significant time commitment from participants. All parents are asked to complete three online questionnaire batteries (1 hour each) and to accompany their child to three child-directed assessment sessions (2 hours each), totalling approximately 9 hours. A subsample of parents (i.e., intervention group) will participate in six intervention sessions (12 hours in total), with additional practice of the techniques required at home. This time commitment may influence which parents choose to participate, potentially resulting in a recruitment bias. Specifically, it may only be parents interested in the topic area who have access to the internet and/or sufficient time who are able to participate in the research.

To address these issues, we will ensure that each online survey is accessible by participants on multiple occasions over the course of 2 weeks. This will allow the questionnaires to be completed in smaller, more manageable sections. To minimize attrition, an email reminder will be sent to the parents once their questionnaires have been active for a week. A monetary incentive will also be provided at each timepoint for those who complete both the child-direct assessments and parent questionnaires. To further illuminate factors which contribute to attrition, we will compare the demographic information of parents who complete the study with those who leave.

**No treatment control group.** The control group in this research study do not receive any form of intervention. Although all parents are free to withdraw at any time, the control group may experience higher levels of withdrawal due to (a) parents being unaware of the importance of the control group, and instead feeling they are not contributing to the research; or (b) parents being specifically interested in participating in the CLK-CUES program. To address these possibilities, the information material will clearly state that the intervention sessions are not a clinical treatment program for anxiety as they are yet to be fully tested in parents of children on the spectrum. The material will also explain how both the intervention and control groups are equally important to the research, as comparisons between these groups can enhance understanding of the impact of anxiety in autism.

Although the use of a wait-list control or the provision of a secondary intervention was considered for parents in the control condition, these strategies were deemed inappropriate. By

the end of T3 the children will exceed the target age-range for CLK, thus rendering the wait-list control unacceptable. Second, an alternative intervention could have an indirect effect on the children's anxiety level, which may have implications for the critical experimental manipulation of anxiety. Given these limitations, it was decided that the control group would receive no treatment as this increases our ability to address the pertinent questions of whether anxiety is related to academic behaviours, and whether the CLK-CUES program is associated with a reduction (or prevention) of anxiety in children on the spectrum. It is important to note, however, that parents will be able to access outside services to assist with their child's anxiety. This is true of parents in both the intervention and control groups. Data will be collected at each timepoint regarding any services accessed to assist with the management of anxiety.

## Ethics and dissemination

Ethical clearance has been provided by Griffith University Human Research Ethics Committee (ref 2019/989) and has approval from the AEIOU Research and Innovation Committee board. Ethical approval will be sought from relevant educational departments once children transition. Any protocol modifications will be approved by these committees and then communicated to participants. All parents will provide informed consent for their child to participate. Parents will also consent to the teachers completing questionnaires about their child before the teacher is approached and asked to consent to participate in the research. We will also seek assent from the children prior to conducting any child-directed assessments.

Although there are no anticipated risks of participation in this research, some parents may become distressed when answering questions about their child's autism or the associated difficulties. If this occurs, the research team can be contacted to discuss the questionnaire and, where necessary, can refer parents to specialised services. Additionally, as this is the first controlled implementation of the CLK-CUES intervention for children on the spectrum, its influence on the children's behaviour is yet to be determined. If parents should become concerned about any changes in their child's behaviour, they will be encouraged to speak with a member of the research team or a healthcare professional. Any adverse effects or unintended consequences will be immediately reported to the HREC ethics committee.

Data will be stored in line with the National Health Medical Research Council best practice [62] and will only be accessible to members of the research team. Data will be anonymised by allocating each participant a unique person identification code. The anonymised data will be saved on a password-protected secure computer drive or in a locked filing cabinet. Any personal identifiable information collected from participants (e.g., consent forms) will be stored securely in a separate location to the deidentified participant data. The dataset will remain with the PI at the host University and all co-investigators will have the opportunity to contribute to the results paper. Upon completion of the final publication from the project, the data will be stored in Griffith University's Research Data Repository, which is accessible and searchable through a web interface.

The overall findings of this research will be communicated via published articles in peer-reviewed journals, conference presentations, and summary reports issued to participating parents and early intervention centres. To maintain anonymity, we will refer primarily to aggregated data rather than to individual responses. Parents will also receive a written summary of their child's results at the end of each assessment stage (Time 1, 2, and 3). This summary can be shared by parents with healthcare professionals. If concerns are noted with regards to anxiety or behaviour scores, an additional statement will be added to the report to encourage parents to share the results with relevant professionals.

## Significance and outlook

This study will make important contributions to the literature examining the role of anxiety in autism. In a methodological advance, anxiety will be experimentally manipulated using the CLK-CUES program. This will capture a range of anxiety levels, thus enhancing the scope and generalisability of the results. These findings will address whether anxiety moderates the relation between autism characteristics and academic learning behaviours, which can subsequently inform the development of interventions aimed at children on the spectrum entering their first school year.

Furthermore, as a secondary outcome, the check for experimental manipulation of anxiety will provide initial insight into the efficacy of the CLK-CUES program on the reduction (or prevention) of anxiety in autism. Until now, CLK has focused on threat expectancy, yet it is hypothesised that incorporating elements relating to the tolerance of uncertainty will provide a more robust intervention (although the current research design cannot test this hypothesis). By examining anxiety at two timepoints post-manipulation (i.e., T2 and T3) we are able to identify both the short-term and longer term impact of the CLK program. This also allows for the early developmental trajectory of anxiety (and its relation to academic behaviours) to be ascertained.

Study results will assist in informing clinicians, educators, children and their families about the influence of anxiety on academic skills. By being aware of the impact of anxiety on academic abilities, appropriate interventions can be selected to minimise/prevent anxiety and support the development of academic skills in young children on the spectrum. Moreover, should the experimental manipulation of anxiety via the CLK-CUES program be effective, this can be used as pilot data to provide the impetus for future large-scale clinical trials to better determine the efficacy of this program and its constituent components. Such larger trials could also then account for other co-occurring conditions with may impact upo academic outcomes and/or interact with anxiety in children on the autism spectrum.

## Summary and conclusion

Anxiety disorders and sub-clinical levels of anxiety are prevalent within autism [13,14]. As anxiety is a key barrier to education in typically developing children [20], it is important to enhance our understanding of this relation for children on the spectrum. This study will collect data that examines the potential impact of anxiety on the academic enablers and academic achievement of young children on the spectrum (4- to 5-year-olds). The findings of this study will assist clinicians and educators in understanding how anxiety impacts academic performance, and will therefore inform the selection of strategies to reduce its impact.

## Supporting information

**S1 Checklist. Reporting checklist for protocol of a clinical trial.**
(DOCX)

**S1 File.**
(PDF)

**S2 File.**
(PDF)

## Author Contributions

**Conceptualization:** Dawn Adams, Kate Simpson, Deb Keen.

**Formal analysis:** Stephanie Malone.

**Funding acquisition:** Dawn Adams, Kate Simpson, Madonna Tucker, Ron M. Rapee, Jacqui Rodgers, Deb Keen.

**Investigation:** Dawn Adams, Stephanie Malone, Kate Simpson, Madonna Tucker, Ron M. Rapee, Jacqui Rodgers, Deb Keen.

**Methodology:** Dawn Adams, Kate Simpson.

**Project administration:** Dawn Adams, Stephanie Malone.

**Resources:** Madonna Tucker.

**Writing – original draft:** Dawn Adams, Stephanie Malone, Ron M. Rapee, Jacqui Rodgers, Deb Keen.

**Writing – review & editing:** Dawn Adams, Stephanie Malone, Kate Simpson, Madonna Tucker, Ron M. Rapee, Jacqui Rodgers, Deb Keen.

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
