## [Decision Letter · Decision Letter 0]

26 Apr 2021

PONE-D-20-28228

Protocol for a longitudinal study investigating the role of anxiety in the development of academic enablers in children on the autism spectrum

PLOS ONE

Dear Dr. Adams,

Thank you for submitting your manuscript to PLOS ONE. After careful consideration, we feel that it has merit but does not fully meet PLOS ONE’s publication criteria as it currently stands. Therefore, we invite you to submit a revised version of the manuscript that addresses the points raised during the review process.

Please make sure you address the statistical comments.

We look forward to receiving your revised manuscript.

Kind regards,

Tarek K Rajji

Academic Editor

PLOS ONE

Journal Requirements:

2. Please include the clinical trial registration information in your Methods section.

[I have read the journal's policy and the authors of this manuscript have the following competing interests:Professor Ron Rapee developed the intervention (Cool Little Kids) that is being used in the study.  th].

Reviewers' comments:

Reviewer's Responses to Questions

**Comments to the Author**

1. Does the manuscript provide a valid rationale for the proposed study, with clearly identified and justified research questions?

Reviewer #1: Yes

Reviewer #2: Yes

Reviewer #3: Partly

Reviewer #4: Yes

2. Is the protocol technically sound and planned in a manner that will lead to a meaningful outcome and allow testing the stated hypotheses?

Reviewer #1: Yes

Reviewer #2: Yes

Reviewer #3: No

Reviewer #4: Yes

3. Is the methodology feasible and described in sufficient detail to allow the work to be replicable?

Reviewer #1: Yes

Reviewer #2: Yes

Reviewer #3: No

Reviewer #4: Yes

4. Have the authors described where all data underlying the findings will be made available when the study is complete?

Reviewer #1: Yes

Reviewer #2: Yes

Reviewer #3: Yes

Reviewer #4: Yes

5. Is the manuscript presented in an intelligible fashion and written in standard English?

Reviewer #1: Yes

Reviewer #2: Yes

Reviewer #3: Yes

Reviewer #4: Yes

6. Review Comments to the Author

You may also provide optional suggestions and comments to authors that they might find helpful in planning their study.

Reviewer #1: I appreciate this opportunity to review this nicely planned research proposal. I would like to apologize for the late reply. I have some comments and suggestions before fully endorsing the publication of this proposal.

1) On Page 6, the authors directly stated “One plausible hypothesis is that anxiety….” I think the publication of this proposal is also of educational purpose, it thus is worth a short mentioning of other potential contributors, e.g. executive dysfunction and motivation problem, etc.

2) On Page 8, “In summary, it is likely that heightened anxiety…..” I appreciate the novel endeavor to link anxiety and academic enabler issues associated with children with ASD. However, there still are numerous other contributing factors to the problems in academic enablers, as mentioned in the last points. Therefore, I suggest that the summary sentences of Introduction should be toned down a bit to something like, “it is likely that heightened anxiety, at least partially, interacts with….”

3) On Page 9, Aim and hypotheses: I find that the modified CLK actually is based on the CLK and further integrates the element of CUES, as explained in the preceding paragraph. However, I don’t find the reason why only the CLK is highlighted in the description of the distinct intervention herein.

4) On Page 10, as the Preparatory Year should be a very Aussie-specific system, it is worth a concise introduction of this system to the non-Aussie readers.

5) Would the interindividual differences in early intervention impact the results? This element seems not to be taken into account in the data analysis part. Similarly, the interindividual differences in receiving services assisting with anxiety are not accounted for in the data analysis model.

6) It seems that this study does not exclude autistic children with co-occurring intellectual impairments. I think this practice is highly laudable. However, I wonder whether intellectual levels would impact the features of academic enablers, and the assigned school options, where the teachers might estimate the academic enablers differently, considering the different expectations for those with co-occurring intellectual disability.

7) On Page 13, the authors explicitly stated that this trial should not be considered a clinical trial. To me, this statement is inaccurate, as the secondary outcome actually involves the changes of anxiety after the CLK+CUES. This statement also contradicts with the discussion in the following paragraph, in which the potential significance in clinical improvement from the intervention is highlighted. Furthermore, as the authors highlighted, there has yet to be data on the CLK or CUES efficacy on anxiety in autism. The results of the current proposal certainly could generate the pilot data for the next-stage clinical trial.

Reviewer #2: Suggest a revision in order to simplify the protocol

As requested by PLoS, I have focussed my review on the more statistical aspects of this protocol. The protocol clearly describes an important research question, it is well written and seems to address all the key issues. However, my main concern, see also some specific points below, is the proposed complex and multi-faceted analysis described (Page 19). Further I suspect the sample size eventually achieved may not be sufficient to justify the analytical methods proposed. My general suggestion is therefore to simplify the protocol as much as possible and thereby reduce the amount of data collected so as to focus on key questions. Otherwise, I fear that there is a danger that the trial will fail for logistical reasons (too much missing data) and the key questions may not be answered. In brief, overloading the parents/children and the investigating team will diminish the chance of a successful completion of this randomised trial.

Specific points

1. As this is a ‘randomised trial’ I suggest this term replaces ‘study’ in the title.

2. Endpoints (Pages 9 and 10): The aims of the trial are not expressed as evaluating the effectiveness of the Anxiety Reduction/Prevention program (ARP) (a combination of CLK and CLUES) as compared to a non-intervention Control at 2-years post baseline (T3). Although this anticipated difference does form the basis of the sample size calculation (Page 11). Neither do they clearly state the aim in relation to the assessment at time T2.

3. Recruitment and Randomisation (Page 10): The protocol seems to suggest that there may be considerable delay between randomisation and the commencement of the intervention. In general terms, to avoid losses (parents/children changing their minds about participation) it is best to start the intervention (whether ARP or Control) as soon as possible after randomisation. Similarly baseline values should be recorded immediately prior to the randomisation. However, I am not sure of the logistical problems here.

4. Randomisation (Page 12): More detail of the randomisation process is required. This Includes: Who generates the list (with details of block size). Also, who holds the list (someone independent of the investigators) and how is the randomisation carried out (by phone to as statistical office?).

5. Sample size (Page 10): For the primary endpoint of the Academic Competence Evaluation Scales (ACES) at 2-years (Is this correct?) post-randomisation the standardised effect size, d = 0.75, then with test size 5%, power 80%, attrition rate (10%) I can confirm the sample size of N = 64 children. However, it would be useful if a direct (statistical) reference is given to the calculation method utilised.

6. Primary analysis: The anticipated attrition implies that only 58 children may be available for analysis. Although this is sufficient for the planned t-test (the phrase ‘and non-centrality parameter’ is not relevant here), I suggest that some of the further analyses, such as hierarchical multiple regression (page 11), will not be very sensible with such numbers. Neither does this method seem relevant to answering the main research question (see Point 2 above). This analysis requires the estimate of a difference between two means via the t-test, with the corresponding confidence interval. In addition, linear regression including intervention and the baseline assessment ACES may be useful.

6. Secondary analysis (page 19, last paragraph): Surely all that is needed here is a similar approach to Point 5 above. I don’t anticipate ANOVA would be required.

Reviewer #3: In this longitudinal experimental study protocol, authors aim to examine- a) the association between anxiety and academic learning behaviours, and b) whether anxiety moderates the association between the core behavioural characteristics of autism and academic learning behaviours in 64 children with autism aged 4-5 years. In this protocol, children will be randomized to receive either a modified Cool Little Kids (CLK) intervention (the combination of CLK and Coping with Uncertainty in Everyday Situations or CUES) (n=32) or no intervention/treatment as usual (n=32). Data collection will be completed at 3 different time points - at baseline (T1), at the end of a 6-month long intervention that parents will receive (T2) and at around 6 months of children transitioning to formal schooling (T3). Research team collecting data and teachers will remain blind. The primary outcome measure appears to be the ? teacher-administered Academic Competence Evaluation Scales assessing academic and enabling behaviours, and the secondary outcome measures are children academic attainment and parents and teacher-rated anxiety scales. This is not classified as a clinical trial since the primary outcome is not a health outcome. This study protocol has received appropriate IRB approval. One of the study authors declared a COI as the author (RMR) previously developed the CLK program.

Strengths:

This type of study examining the role of anxiety on the academic performance and learning in autistic children is much needed as they may pave the way to the successful development of school-based intervention that can improve outcomes in autistic children.

Use of parent- and teacher-rated measures, use of multiple measures including sensory, behaviour-related, cognitive and adaptive scales.

Carefully designed intervention, taking into consideration some common potential pitfalls.

Having an author from the population health sciences institute.

Limitations:

In my opinion, there are several limitations of this protocol:

I am not clear about the exact primary measure of this protocol. In the abstract and data analysis section Academic Competence Evaluation Scales - teacher rated form was mentioned as primary but in the table Wechsler individual achievement test was mentioned as the primary.

It would be easier to interpret if authors could add a figure to describe the study design.

Authors mentioned that there was no identified predictor of academic underperformance in autistic children. They reviewed literature to support their focus on anxiety in autistic children. However, this link appears non-specific as they also mentioned that anxiety is a strong predictor of academic underperformance in neurotypical children as well. Furthermore, given that clinically manifested anxiety is seen in 4 out of 10 autistic children, how would they justify the distribution of anxiety between the two groups? Would there be any value in having a neurotypical control group with anxiety? How about autistic children with and without significant anxiety to control for non-anxiety related autism factors contributing?

The authors included potential moderators (e.g. parental mental health, etc) in the analysis but they did not review information on these potential moderators related to autism in the introduction.

There are no exclusion criteria - what about unstable or uncontrolled medical conditions interfering with the academic ability, eg, epilepsy (common in autistic children), specific learning disability, intellectual disability, speech problem, genetic disorders that affect mobility and communication, presence of co-occuring mental health conditions such as attention-deficit hyperactivity disorder that affects academic performance. The design, if focused on anxiety as a mediator in the context of ‘autism’, should consider all these potential confounders.

Authors mentioned effect size calculation to justify the sample size but it is not clear if they referred to Cohen’s d and why did they use t-statistic when the proposed analysis plan includes multiple hierarchical regression, mixed ANOVAs.

The process of blinding is not also clear to me, for instance, authors mentioned that the research team members collecting data will be blind. However, the CLK intervention will be administered by a trained member of the research team and a second research team member will assess the validity. A separate team of analysts/researchers need to deliver the intervention. Also, will the data analysts be blind?

In the data analysis plan, authors need to include details regarding controlling for multiple comparisons. The sample size should reflect that as well but a sample size of 64 is not sufficiently justified.

The authors will collect information on sensory profiles, adaptive, etc but how they are going to handle all these additional measures in the analysis plan? It remains unclear.

I did not see any sex/gender consideration, which is a potentially powerful mediator in autism.

Reviewer #4: Thank you for the opportunity for reviewing this manuscript.

This is a research protocol investigating the relation between anxiety and academic achievement in children on the autism spectrum. This randomized longitudinal study aims to study the moderating effects of anxiety in development of academic enablers in children with autism spectrum disorder. Secondary outcome measures of the study are academic attainment and anxiety. The proposed study has a randomised, longitudinal design and a sample size of 64, based on appropriate sample size calculation. A battery of parent-reported measures and academic performance measures are proposed to test the hypothesis. The authors have used a robust methodology for this study and have addressed issues related to recruitment bias and waitlist control in the study. Use of SPIRIT checklist add rigour to the quality and transparency of the study protocol. This methodologically sound study can contribute to current understanding of the role of anxiety in autism and has the potential to inform development of interventions in children with autism in their early school years.

The authors of the study protocol clearly state the rationale for the study with clearly defined research questions. They also state that the hypothesis of heightened anxiety interacting with academic enablers to contribute to poor academic performance of children on the spectrum has not been tested to date. In this study, a modified anxiety reduction/prevention program comprising of Cool Kids program (CLK), a parent-led program for anxiety in pre-school children and Coping with Uncertainty in Everyday Situations (CUES), another parent mediated intervention, will be used as an intervention for anxiety. The authors have mentioned about a pilot study on the role of CLK in anxiety intervention. Please describe the modified program and cite additional supporting references, if any, about the role of CLK in moderating anxiety in autism.

CLK has been used for children in the age group 8-15 years and this study is designed for 4–5-year-old children on the spectrum. A pilot study of the modified version has been carried out but it is not clear if the participants were pre-school children. In addition, the validity of the modified anxiety reduction program as an intervention for the participants is not clear. Please address this issue in the research protocol.

7. PLOS authors have the option to publish the peer review history of their article (what does this mean?). If published, this will include your full peer review and any attached files.

Reviewer #1: **Yes: **Hsiang-Yuan Lin

Reviewer #2: **Yes: **David Machin

Reviewer #3: No

Reviewer #4: **Yes: **Anupam Thakur

---

## [Author Response · Author response to Decision Letter 0]

17 Jun 2021

We thank the editor and the reviewers for their feedback on the manuscript. We have attended to each reviewer’s comments in turn in the table uploaded with the manuscript.

---

## [Decision Letter · Decision Letter 1]

27 Aug 2021

Protocol for a longitudinal study investigating the role of anxiety on academic outcomes in children on the autism spectrum

PONE-D-20-28228R1

Dear Dr. Adams,

We’re pleased to inform you that your manuscript has been judged scientifically suitable for publication and will be formally accepted for publication once it meets all outstanding technical requirements.

Kind regards,

Tarek K Rajji

Academic Editor

PLOS ONE

Additional Editor Comments (optional):

Reviewers' comments:

Reviewer's Responses to Questions

**Comments to the Author**

1. Does the manuscript provide a valid rationale for the proposed study, with clearly identified and justified research questions?

Reviewer #1: Yes

Reviewer #2: Yes

Reviewer #4: Yes

2. Is the protocol technically sound and planned in a manner that will lead to a meaningful outcome and allow testing the stated hypotheses?

Reviewer #1: Yes

Reviewer #2: Yes

Reviewer #4: Yes

3. Is the methodology feasible and described in sufficient detail to allow the work to be replicable?

Reviewer #1: Yes

Reviewer #2: Yes

Reviewer #4: Yes

4. Have the authors described where all data underlying the findings will be made available when the study is complete?

Reviewer #1: Yes

Reviewer #2: Yes

Reviewer #4: Yes

5. Is the manuscript presented in an intelligible fashion and written in standard English?

Reviewer #1: Yes

Reviewer #2: Yes

Reviewer #4: Yes

6. Review Comments to the Author

You may also provide optional suggestions and comments to authors that they might find helpful in planning their study.

Reviewer #1: I appreciate that the authors have addressed all my comments adequately. Now I am happy to endorse the publication of this protocol in the current form.

Reviewer #2: No comment as this is the second review but see my review.

Reviewer #4: Thank you for the opportunity to review the manuscript.

The authors have provided rationale for the proposed study as well as clearly identified and justified the research questions. It is a methodologically robust study with attention to detail.

Thank you for addressing the points mentioned in the reviewer's feedback.

7. PLOS authors have the option to publish the peer review history of their article (what does this mean?). If published, this will include your full peer review and any attached files.

Reviewer #1: **Yes: **Hsiang-Yuan Lin

Reviewer #2: **Yes: **David Machin

Reviewer #4: **Yes: **Anupam Thakur

---

## [Editor Report · Acceptance letter]

6 Sep 2021

PONE-D-20-28228R1 

Protocol for a longitudinal study investigating the role of anxiety on academic outcomes in children on the autism spectrum 

Dear Dr. Adams:

I'm pleased to inform you that your manuscript has been deemed suitable for publication in PLOS ONE. Congratulations! Your manuscript is now with our production department. 

Kind regards, 

on behalf of

Dr. Tarek K Rajji 

Academic Editor

PLOS ONE